# Impact of Chill and Heat Exposures under Diverse Climatic Conditions on Peach and Nectarine Flowering Phenology

**DOI:** 10.3390/plants12030584

**Published:** 2023-01-29

**Authors:** Pavlina Drogoudi, Celia M. Cantín, Federica Brandi, Ana Butcaru, José Cos-Terrer, Marcello Cutuli, Stefano Foschi, Alejandro Galindo, Jesus García-Brunton, Eike Luedeling, María Angeles Moreno, Davide Nari, Georgios Pantelidis, Gemma Reig, Valentina Roera, Julien Ruesch, Florin Stanica, Daniela Giovannini

**Affiliations:** 1Department of Deciduous Fruit Trees, Institute of Plant Breeding and Genetic Resources, Hellenic Agricultural Organization (ELGO)-‘DIMITRA’, 59200 Naoussa, Greece; 2Department of Pomology, Estación Experimental de Aula Dei (EEAD), CSIC, 50059 Zaragoza, Spain; 3Research Centre for Olive, Fruit and Citrus Crops, Council for Agricultural Research and Economics (CREA), 47121 Forlì, Italy; 4Research Centre for Study of Food and Agricultural Products Quality, University of Agronomic Sciences and Veterinary Medicine of Bucharest, 011464 Bucharest, Romania; 5Instituto Murciano de Investigación y Desarrollo Agrario (IMIDA), 30150 Murcia, Spain; 6Research Centre for Olive, Fruit and Citrus Crops, Council for Agricultural Research and Economics (CREA), 00134 Rome, Italy; 7Rinova Soc. Coop., 47522 Cesena, Italy; 8Institute of Crop Science and Resource Conservation (INRES)–Horticultural Sciences, University of Bonn, 53121 Bonn, Germany; 9Fondazione ricerca, Innovazione e Sviluppo Tecnologico Dell’agricoltura Piemontese, AGRION, 12030 Manta, Italy; 10Fruitcentre, Fruit Production Department, Institute of Agrifood Research and Technology (IRTA), 25003 Lleida, Spain; 11Centre Technique Interprofessionel des Fruits et Légumes (CTIFL), 30127 Bellegarde, France

**Keywords:** resilience, chilling requirement, heat requirement, *Prunus persica*

## Abstract

The present study aims to generalize cultivar-specific tree phenology responses to winter and spring temperatures and assess the effectiveness of the Tabuenca test and various chill and heat accumulation models in predicting bloom dates for a wide range of climatic conditions and years. To this end, we estimated the dates of rest completion and blooming and correlated them with observed bloom dates for 14 peach and nectarine cultivars that were evaluated in 11 locations across Europe (Greece, France, Italy, Romania and Spain), within the EUFRIN cultivar testing trial network. Chill accumulation varied considerably among the studied sites, ranging from 45 Chill Portions (CP) in Murcia-Torre Pacheco (Spain) to 97–98 CP in Cuneo (Italy) and Bucharest (Romania). Rest completion occurred latest or was not achieved at all for some cultivars in the southern sites in Murcia. Dormancy release happened earliest in Bucharest and Cuneo, sites where heat accumulation had a strong influence on the regulation of bloom time. Blooming occurred earliest in the moderately cold regions of Lleida (Spain) and Bellegarde (France), and 7–11 days later in the warmer locations of Rome (Italy) and Naoussa (Greece), suggesting that bloom timing is strongly influenced by delayed rest completion in these locations. The Dynamic Model resulted in both more homogeneous chill accumulation across years and better predictions of bloom dates, compared with the Utah, Positive Utah and Chilling Hours models. Prediction of bloom dates was less successful for low-chill cultivars than for medium- and high-chill cultivars. Further climatic and experimental data are needed to make estimates of the climatic needs of peach cultivars more robust and to generate reliable advice for enhancing the resilience of peach production under varying and changing climatic conditions.

## 1. Introduction

Like other temperate fruit and nut trees, peach trees require adequate winter chill before they can produce economically viable yields. Cultivar-specific chilling requirements must be fulfilled before trees become receptive to heat, which ultimately allows them to flower and eventually bear fruit. The chilling requirement is thus a major determinant of where a certain species or cultivar can be grown. Peach is sensitive to temperature, with cultivars showing considerable variation in chilling requirements [1,2]. Available estimates of chilling requirements have been difficult to compare, due to differences in the response of studied cultivars to different climates, use of unreliable chill calculation models or variation in the methodology used to determine the trees’ agroclimatic needs. Insufficient chill exposure may result in erratic, prolonged and delayed bloom, low yield, and aberrant peach fruit shape (elongated fruits with protruding tips) [3]. Prolonged winter chill accumulation, in contrast, may reduce the heat requirement in spring [4,5].

Predictions of tree responses are hampered by a lack of knowledge on the elasticity of peach cultivars to variation in chill and heat exposure. In a study by Li et al. [6], long-term bloom observations of peach cultivars in China showed a dramatic advance of 11.1 days between 1982 and 2012. Bloom date was shown to be negatively correlated with the average air temperature from February to April indicating that higher temperatures in this critical period accelerated developmental processes, leading to early occurrence of spring events. In warmer regions, by contrast, a delay of spring events may occur as a result of later fulfillment of the chilling requirement, as has been demonstrated, for instance, for pistachios and almonds in Tunisia [7,8]. The identification of resilient cultivars with good adaptability to variation in climate is one of the most important strategies to counteract the adverse impacts of climate change. 

The most widely used method for estimating chilling requirements is the Tabuenca test [9]. In this test, tree shoots are collected from the field on multiple occasions during the winter period, after receiving increasing levels of chill exposure. The shoots are then forced in a growth chamber to assess the ability of their buds to develop into flowers. The results allow insights into the timing of endodormancy release. Chilling requirement estimates with the Tabuenca test have been variable, showing strong dependence on year and climatic conditions. Higher chilling requirements have been determined in apricot [10] and cherry [11,12] cultivars grown in cooler areas compared to warmer regions. The consistency of chilling requirement estimates with the Tabuenca test was also tested in peach cultivars in Lleida (Spain) and Naoussa (Greece) [13]. Results showed that the site did not affect the chill or heat requirements of the studied cultivars when the Dynamic Model (estimates in Chill Portions; CP) and the Utah Model (estimates in Chill Units; CU) were used. When using the Positive Utah Model (Positive Chill Units; PCU) and the Chilling Hours Model (CH), however, the studied cultivars showed higher chilling requirements in Lleida than in Naoussa. 

A network of cultivar testing trials was established in the European Union (EU) in 2018 under the EUFRIN (European Fruit Research Institutes Network) Apricot and Peach working group (https://eufrin.eu/working-groups/apricot-and-peach, accessed on 26 January 2023) which aimed to study the influence of climatic conditions on peach phenology, adaptation and fruit traits [14]. The trials include sites across a wide gradient of climatic conditions, ranging from very warm conditions in southern Spain to moderately cold conditions in France, northern Spain, Italy and Greece, and cold regions in northern Italy and Romania. These locations span a wide range of chill and heat accumulation during the chilling and forcing periods preceding bloom. The harmonized multi-site datasets will allow assessments of Genotype x Environment interactions for traits related to bloom timing, productivity and fruit quality. They will also help identify cultivars that are particularly vulnerable or robust to the impacts of climatic variation and change. Chill and heat requirements are considered useful indicators for predicting the viability of deciduous fruit species, or their specific cultivars, in locations where they have not yet been cultivated. However, deriving reliable estimates of these climatic needs requires standardized observations across multiple locations according to harmonized protocols. For peach, such data have been elusive so far. This study reports on efforts by the EUFRIN network to close this knowledge gap. 

The aims of the present research were to (a) study the impacts of different levels of chill and heat exposure during the chilling and forcing periods on the phenological stages of a heterogeneous group of peach and nectarine cultivars, and (b) assess the effectiveness of the Tabuenca test and different chill and heat accumulation models to predict bloom dates under different pedoclimatic conditions and in different years. 

## 2. Materials and Methods

### 2.1. Plant Material and Experimental Conditions 

The study was conducted on 14 peach [*Prunus persica* (L.) Batsch] and nectarine [*Prunus persica* (L.) Batsch var. nucipersica] cultivars, belonging to the collaborative EUFRIN cultivar testing trial. The cultivars were chosen for being well known, with proven high and constant yield and fruit quality, representing the most important commercial peach fruit categories and with diverse genetic backgrounds (Table 1) [14]. The trials were established in 2018 in 11 locations: Bellegarde in France; Naoussa in Greece; Cuneo, Forlì, Rome and Tebano in Italy; Bucharest in Romania; and Lleida, Murcia-Yéchar (YE), Murcia-Torre Pacheco (TP) and Zaragoza in Spain. The geographical locations, tree densities and training systems used at each site are presented in Table 2. 

Trees were managed according to an integrated management system. The experiments were conducted during three consecutive years (2020–2022). 

### 2.2. Bloom Characterization

At all sites, we recorded data on flowering phenology for all cultivars, focusing on three phenological stages: (1) the beginning of bloom, defined as the date when about 10% of flowers are open; (2) full bloom, the date when 80% of the flowers are open: and (3) the end of bloom, the date when 90% of petals have fallen [corresponding to stages 61, 66 and 67 of the BBCH scale, respectively [15]. All dates were expressed as Julian dates (JD; the day of the year). Bloom duration was calculated as the interval between the beginning and the end of blooming.

### 2.3. Chill and Heat Models 

In each field trial, hourly temperatures were recorded from October to April during each year of the three-year period (2020–2022) by automatic meteorological stations. From these hourly records, we computed daily mean temperature (T_mean_). 

Chill accumulation for the period October to February was calculated according to four models—the Dynamic Model, the Utah Model, the Positive Utah Model and the Chilling Hours Model—using hourly temperature values as input. 

The Dynamic Model [16,17] performed better than the other models in quantifying the chilling requirements of several cultivars in a study carried out in Lleida (Spain) and Naoussa (Greece) [13]. This model postulates that winter chill accumulates in a two-step process. During the first step, cold temperatures lead to the formation of a precursor of the dormancy-breaking factor. Once a certain quantity of this precursor has accumulated, it can be transformed into a Chill Portion (CP) by a process that occurs most efficiently at moderate temperatures.

The second model we tested was the Utah Model, which quantifies accumulated chill in ‘Chill Units’ (CU). It assigns different weights to different temperature ranges, including negative weights for temperatures > 16 °C [18]. Evidence from warm growing regions has indicated, however, that the Utah Model may be exaggerating the chill negation effect. In consequence, the Utah Model has been adapted by limiting or ignoring chill negation. One of the modified versions is the Positive Utah Model, which was created by removing the negative contributions of warm temperatures from the original equation of the Utah Model. Each hour is treated independently and allocated between 0 and 1 ‘positive chill units’ (PCU), according to the prevalent temperature. All the PCUs are summed up for the seasonal total [19]. 

We also evaluated the performance of the Chilling Hours Model (Hutchins 1932, cited by Weinberger [20]), which calculates the accumulated ‘chill hours’ (CH) as the number of hours with temperatures between 0 and 7.2 °C. Even though the historic origins of the Chilling Hours Model are obscure, it is hardly biologically plausible and it has consistently performed poorly [21], the Chilling Hours Model is still widely used to describe the chilling requirement since it is easy to understand and apply [22]. 

Heat accumulation was estimated by using the model proposed by Richardson et al. [18], which expresses heat in growing degree hours (GDH). According to this model, heat builds up when hourly temperatures range between 4.5 and 36 °C (at different rates depending on the temperature), with maximum accumulation at an optimal temperature of 25 °C. 

### 2.4. Estimated Dates of Dormancy Release and Bloom

To estimate the date of dormancy release, we relied on cultivar-specific chilling requirements, previously determined by simultaneously conducted Tabuenca tests in Lleida (Spain) and Naoussa (Greece) using the Dynamic, Utah, Positive Utah and Chilling Hours models [13]. Chill requirements estimated with the Dynamic Model were 48 CP for ‘Carla’, 51 CP for ‘Patty’, 56 CP for ‘Big Top’ and ‘Emeraude’, 58 CP for ‘Elegant Lady’, ‘Flatstar’, ‘Sweetregal’ and ‘Venus’, 61 CP for ‘Gladys’ and ‘Sweet Dream’, 62 CP for ‘Sweet Cap’ and ‘Nectarperf’, 64 CP for ‘Catherina’ and 69 CP for ‘O’Henry’, respectively. Following the date of dormancy release, bloom dates were estimated as the day when the heat requirement for each cultivar was reached. According to Pantelidis et al. [13], heat requirements varied between 3524 and 4672 × 10^1^ GDH for the period between the end of endodormancy and the end of bloom. 

Since not all cultivars are established in all locations and some temperature data could not be retrieved, the number of observed bloom dates we obtained differed across cultivars: 26 for ‘Carla’; 25 for ‘Elegant Lady’; 24 for ‘Big Top’ and ‘Sweet Dream’; 23 for ‘Sweet Cap’ and ‘Patty’, 22 for ‘Nectarperf’; 21 for ‘Catherina’ and ‘Emeraude’; 20 for ‘Gladys’; 18 for ‘Flatstar’; 16 for ‘Venus’; 15 for ‘Sweetregal’ and 14 for ‘O’Henry’, respectively.

### 2.5. Statistical Analyses and Model Validation 

Results on the date of dormancy release and bloom data were subjected to a two-way analysis of variance using site and cultivar as treatments, while observations in different years were treated as replicates. For the date of dormancy release, analysis of variance (ANOVA) was applied for 11 sites, 14 cultivars and three replicate years. For bloom data, ANOVA was applied for 9 locations (Bellegarde, Bucharest, Cuneo, Forlì, Lleida, Naoussa, Rome, Tebano and Murcia-YE), 7 cultivars (‘Big Top’, ‘Carla’, ‘Catherina’, ‘Elegant Lady’, ‘Nectarperf’ and ‘Sweet Dream’) and for 2021 and 2022. Significant differences among individual means were determined using the Tukey multiple range test, with ANOVA F-test results accepted as significant at *p* < 0.05. Statistical analyses were performed using SPSS, version 13.0 (SPSS Inc., Chicago, IL, USA). 

We also calculated mean values and coefficients of variation (CV%) for estimates of chill and heat requirements. 

Model performance for the observed and calculated full bloom dates estimated using the Dynamic, Utah, Positive Utah and Chilling Hour models was validated using the commonly used root mean square error of the prediction (RMSE). A disadvantage of the RMSE is that it does not incorporate information on the variation among the observed values, which is important for evaluating model fit. To account for this, we also computed the ratio of performance to interquartile distance (RPIQ), for which the interquartile distance (75th percentile minus 25th percentile) is also calculated and divided by the RMSE [23,24]. The RPIQ takes account of both the prediction error and variation of observed values, and is therefore a more objective indicator than the RMSE, facilitating comparisons of model performance across different geographic contexts. The greater the RPIQ, the stronger is the predictive capacity of the model.

## 3. Results and Discussion

### 3.1. Chill and Heat Accumulation 

During the periods from October to March in the study seasons of 2019–2020, 2020–2021 and 2021–2022, January was the coldest month, followed by December and February (Figure 1). Mean monthly T_mean_ between October and March was lowest in Bucharest and Cuneo (6.2–6.3 °C), followed by Tebano, Forlì, Lleida and Zaragoza (8.4–9.5 °C) and Bellegarde, Naoussa and Rome (10.0–10.2 °C). The warmest sites were located in Murcia (YE and TP, 13.0 °C and 13.8 °C, respectively). The mean of monthly T_mean_ across all months of the dormancy season was highest in 2019–2020 (10.7 °C), compared with 2020–2021 and 2021–2022 (9.5 and 9.1 °C, respectively). Differences between years were largest in Naoussa, Forlì, Rome, Bellegarde and Lleida (2.0, 1.9, 1.5, 1.4 and 1.3 °C, respectively), and smallest in Zaragoza, Murcia-TP and Murcia-YE (0.9, 0.6 and 0.5 °C, respectively).

We evaluated the progress of chill accumulation and final chill across eleven European sites with importance to peach cultivation. We quantified chill accumulation between the beginning of October and the end of February using four chill models (Figure 2). Chill accumulation clearly varied between the studied locations. Significant chill usually started to accumulate around the beginning of November in Bucharest and Cuneo, 15 days later in Tebano, Forlì, Lleida, Zaragoza and Bellegarde, 30 days later in Naoussa and Rome, and more than 60 days later (in January) in Murcia-YE and Murcia-TP. 

According to the Dynamic Model, chill accumulation was highest in Bucharest, followed by Cuneo, Tebano, Forlì, Lleida, Zaragoza, Bellegarde, Rome, Naoussa, Murcia-YE and Murcia-TP, with total accumulation ranging between 45 and 97 CP. According to the Utah (CU), Positive Utah (PCU) and Chilling Hours (CH) models, the sites mostly followed a similar order, with chill totals ranging from 639 to 1985 CU, from 1177 to 2080 PCU and from 350 to 1836 CH, respectively. The coefficient of variation (CV%) was greatest for the Chilling Hours Model, followed by the Utah Model, the Dynamic Model and the Positive Utah Model (CV% = 39.8, 30.1, 22.8 and 18.1, respectively). The hard temperature threshold at 7.2 °C in the Chilling Hours Model, above which chill effectiveness ceases abruptly, and the chill negations in the Utah model may be the reasons for the high variation found among estimates of chilling requirements produced by the CH and CU models, especially in mild-weather situations, such as those found in Murcia-TP [25]. 

In general, chill accumulation was lowest in 2019–2020, followed by 2020–2021 and 2021–2022 (means of 72, 80 and 84 CP, respectively), and differences were largest in Rome, Naoussa, Bellegarde and Forlì (13–19 CP difference between 2020 and 2022), reflecting differences in T_mean_ (Figure 1). 

Heat accumulation patterns during the dormancy season varied strongly across the 11 study locations (Figure 3). Total heat accumulation between October and March ranged from 11,110 GDH in Bucharest to 40,550 GDH in Murcia-TP. Heat accumulation generally followed an opposite trend to chill accumulation, with highest values in October and lowest in January. 

### 3.2. Rest Completion 

For all 14 peach and nectarine cultivars, chill and heat requirements for bloom had previously been determined in Lleida (Spain) and Naoussa (Greece) by Pantelidis et al. [13]. According to this study, rest completion was expected after 49 to 69 CP, 920–1321 CU, 1109–1491 PCU and 446–866 CH, while bloom was expected after 3524 to 4672 GDH. In the present study, rest completion was not achieved in high-chill cultivars in some years and sites. For example, in Murcia−YE and Murcia−TP, high-chill cultivars such as ‘Nectarperf’, ‘Catherina’, ‘O’Henry’, ‘Gladys’ and ‘Sweet Dream’ never fulfilled their chilling requirements (Figure 4). Both site and cultivar significantly affected the date of rest completion, with no significant interaction between site and genotype. 

In Murcia−YE and Murcia-TP, average dormancy release was latest (JD 74–78), because of the delay in covering chilling requirements, followed, in order, by Rome, Naoussa, Bellegarde, Zaragoza, Forlì, Lleida, Tebano, Bucharest and Cuneo (JD 42, 27, 21, 19, 14, 12, 9, 3 and 1, respectively, Figure 4). In Naoussa, during the warmest year 2020, dormancy release occurred in March for some cultivars, yet in the cooler years 2021 and 2022, dormancy release was estimated to occur much earlier (in January). 

Following the order of the cultivars’ chilling requirements, rest completion was earliest in ‘Carla’ and ‘Patty’ and latest in the high-chill cultivars ‘O’Henry’, ‘Catherina’ and ‘Nectarperf’ (Figure 4).

### 3.3. Bloom Dates

Beginning of bloom usually occurred latest in the coldest locations, Bucharest (26 March, JD 85), and Cuneo (17 March, JD 77) (Figure 5). Considering that rest completion at these sites was estimated to occur between 15 December and 15 January (Figure 4), the ecodormancy period lasted for more than two months, which we attribute to low heat accumulation in January and February (Figure 3). This result supports earlier evidence indicating that in cold-winter climates, bloom timing is more strongly regulated by heat accumulation, as has been shown for chestnuts grown in Beijing [26], apricots in the United Kingdom [27] and cherries in Germany [28]. 

Bloom occurred earliest in the moderately-cold regions of Lleida and Bellegarde (25 February, JD 55) and latest in the warmer locations of Rome, Naoussa and even Murcia-YE (JD 63 69)) (Figure 5). For Murcia−YE and Murcia−TP, previous work had shown that chill accumulation regulated the time of rest completion. We conclude, therefore, that the delay in bloom occurrence in these locations was caused by late rest completion. Similar findings have been presented for peach bloom in a warm growing region in Argentina [29], and for the leaf emergence of walnuts in California [30]. Results from the present study show that in Rome and Naoussa, temperature increases may endanger the productivity of peach orchards since chill accumulation may become marginal for some cultivars.

‘Carla’ was the earliest to bloom (JD 60, 29 February) (Figure 5), probably due to the early rest completion and the lower chilling requirement (48.3 CP) compared to all other cultivars evaluated in this study. 

### 3.4. Observed vs. Estimated Bloom Dates

To confirm the reliability of estimating bloom dates using the Dynamic, Utah, Positive Utah and Chilling Hours models, we calculated the RMSEs and RPIQs (Figure 6). Root mean square errors were very high and RPIQ were very low for Murcia-YE, suggesting a large discrepancy between expected and observed dates. RMSE values were generally lower and RPIQ values were higher when the Dynamic Model was used for the calculations, followed by the Utah, Positive Utah and Chilling Hours models, lending further weight to earlier evidence indicating the superiority of the Dynamic Model [21].

The models used in the present study performed reasonably well in Bucharest and Bellegarde but appeared poorly suited for Zaragoza. Significant differences between observed and expected bloom dates were documented for the low-chill cultivars ‘Patty’ and ‘Carla’, while the smallest differences were determined for ‘Sweet Dream’, ‘Big Top’ and ‘Catherina’. In a recent study by Sawamura et al. [31], observed bloom dates collected for three peach cultivars and one selection grown in different locations in Japan were compared with predicted bloom dates calculated with the Utah Model. In agreement with our results, the authors found low suitability of this model for the low-chill selection. Moreover, they found higher RMSE values when comparisons were made using data obtained from different locations than for data from one location across many years (3.9–4.6 days vs. 1.8–3.3 days). 

A reason for the relatively poor performance of all prediction models may be that the assumption of chill and heat accumulation occurring strictly sequentially could be an oversimplification. Some evidence suggests that endodormancy and ecodormancy may be overlapping in some species [32], and there may even be an influence of accumulated chill on a tree’s heat requirement [33,34]. 

Further reasons for inaccuracies may be that cultivars may differ in their apparent base temperature and in their heat requirements for floral bud break [35,36]. To a small extent, bloom dates obtained from different locations may also be affected by differences in tree vigor caused by variation in training systems and tree densities, or by other environmental factors, such as water conditions [37,38]. 

## 4. Conclusions

Results indicated that in some locations such as Rome and Naoussa, blooming is strongly regulated by the time of dormancy release. Climate warming may endanger peach productivity in these places since chill accumulation may soon become marginal for some cultivars. Compared with the Utah, Positive Utah and Chilling Hours Models, the Dynamic Model produced both more homogeneous chill accumulation along years and better predictions of bloom dates, suggesting its better suitability for a range of climatically different regions. The prediction of bloom dates was better for medium- to high-chill-requirement cultivars such as ‘Sweet Dream’, ‘Big Top’ and ‘Catherina’, whereas poor predictions were obtained for low-chill cultivars. Further climatic and experimental data are needed to improve the accuracy of the present experimental data and study the potential plasticity of peach cultivars under different climatic conditions.

## Figures and Tables

**Figure 1 plants-12-00584-f001:**
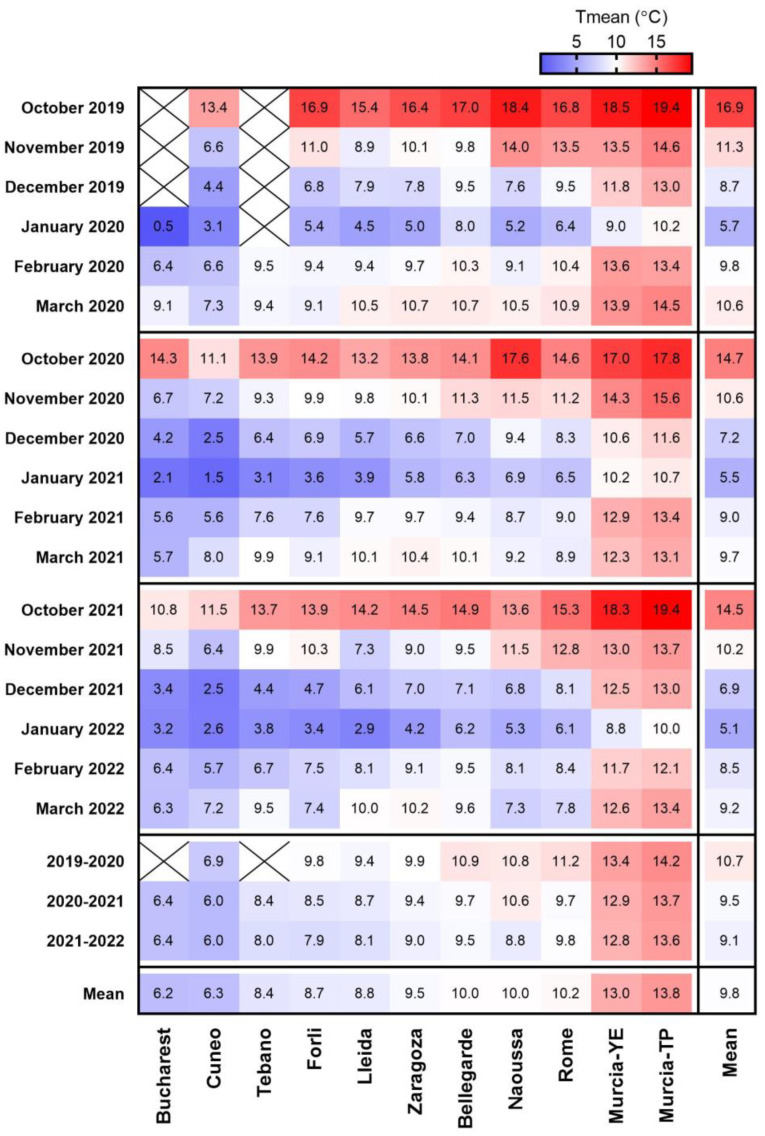
Heat-map presenting monthly mean temperature (°C) during October–March 2019–2022 in 11 European sites, presented from the coolest to the warmest. Mean values of monthly mean temperatures are presented for the above periods in each site and month. X = no data. YE = Yéchar; TP = Torre Pacheco.

**Figure 2 plants-12-00584-f002:**
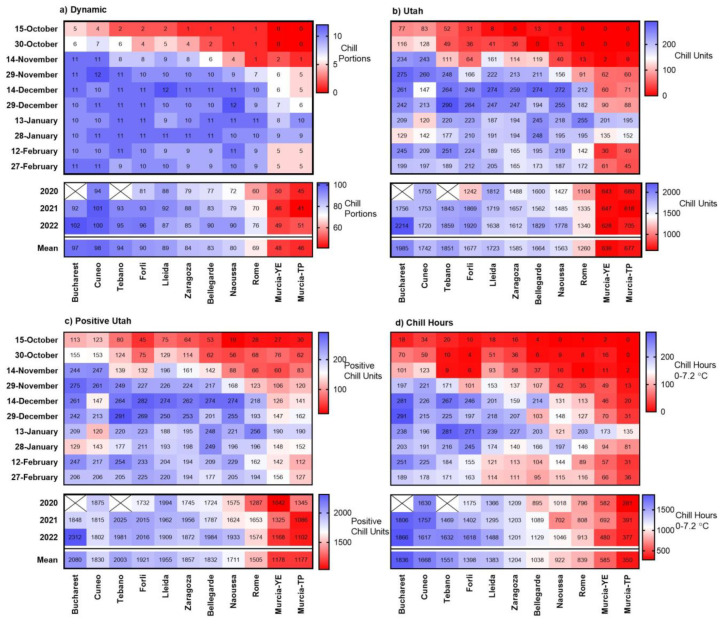
Heat-maps presenting the sum of chill accumulation during 15-d periods (from 1 October to 29 February) and yearly sums (2019–2022) and calculated with the (**a**) Dynamic (Chill Portions), (**b**) Utah (Chill Units), (**c**) Positive Utah (Positive Chill Units) and (**d**) Chill Hours models. X = no data. YE = Yéchar; TP = Torre Pacheco.

**Figure 3 plants-12-00584-f003:**
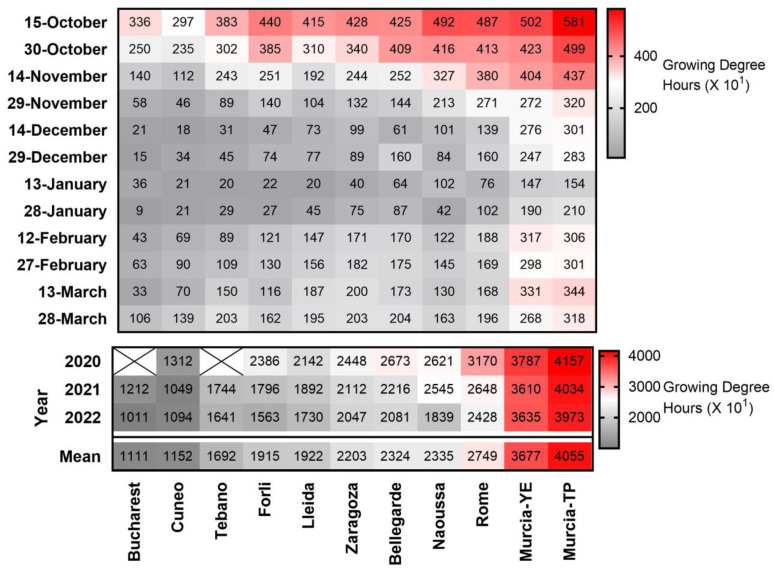
Heat-map presenting heat accumulation (in growing degree hours (GDH) (X10^1^)) accumulated during 15-day periods from 1 October until 29 March in 11 European sites in the 2019–2022 three-year period. The sum of GDH during February–March is also presented for each studied period and year. X = no data. YE = Yéchar; TP = Torre Pacheco.

**Figure 4 plants-12-00584-f004:**
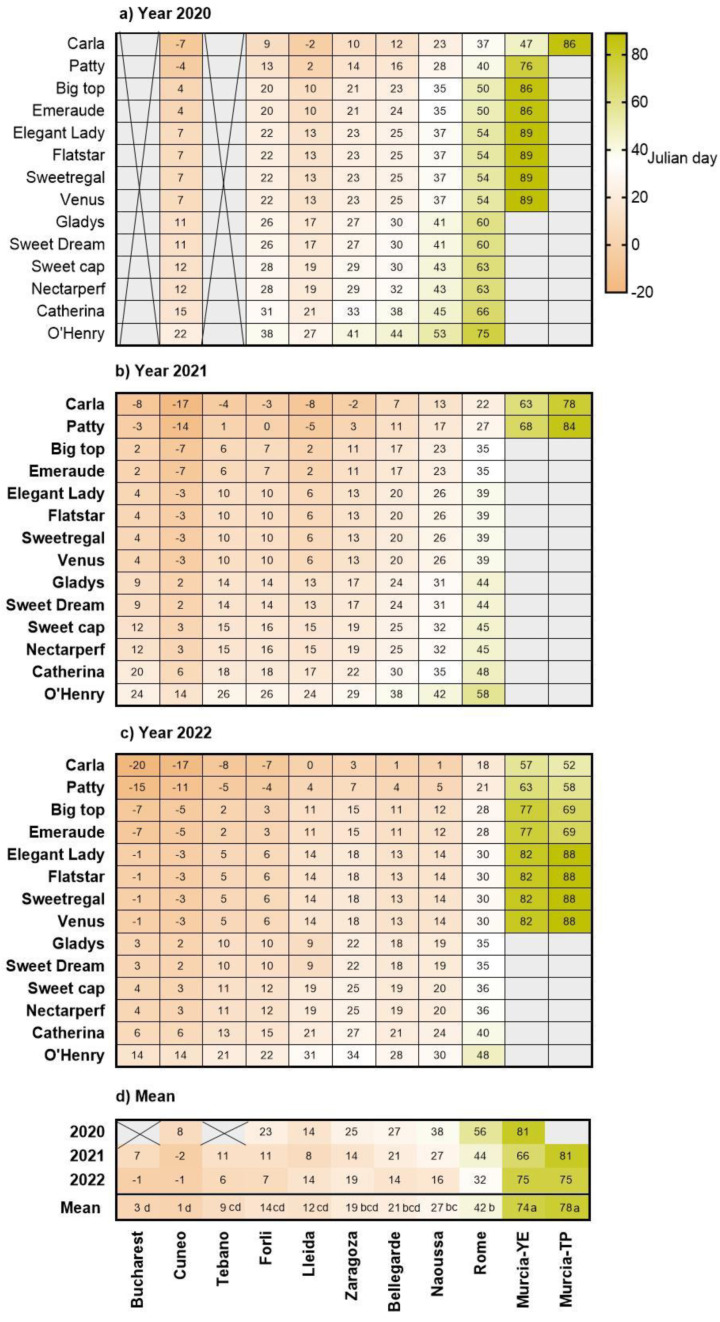
Heat-map showing Julian days of dormancy release, estimated as the day that chill was satisfied, for 14 peach and nectarine cultivars grown in 11 European sites, during (**a**) 2020, (**b**) 2021, (**c**) 2022, and (**d**) mean values. X = no data. Empty cells indicate that dormancy release was not achieved. YE = Yéchar; TP = Torre Pacheco. Negative values indicate days in the previous year. Means with different letters indicate significant differences among sites and cultivars, using the Tukey multiple range test.

**Figure 5 plants-12-00584-f005:**
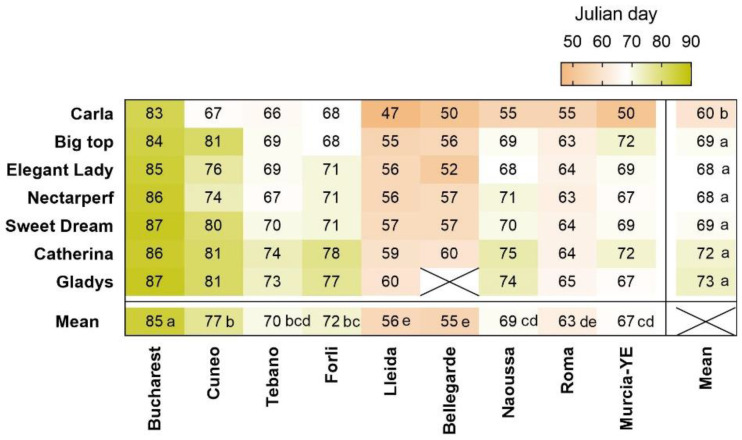
Heat-map showing Julian days of beginning of bloom in 7 cultivars located in 9 sites. Mean values for 2021 and 2022 are shown. X = no data. YE = Yéchar. Means with different letters indicate significant differences among sites and cultivars, using the Tukey multiple range test.

**Figure 6 plants-12-00584-f006:**
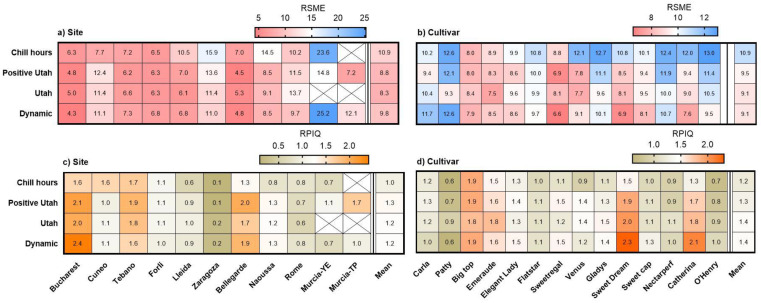
(**a**,**b**) Root mean square error (RMSE), and (**c**,**d**) ratio of performance to interquartile distance (RPIQ), for the observed versus estimated full bloom dates for (**b**,**d**) 14 peach and nectarine cultivars, and (**a**,**c**) at 11 European sites, calculated using four models. Sites are presented from those having the highest to the lowest chill accumulation, and cultivars from having the lowest to the highest chill requirements. Chill requirements are estimated using the Dynamic Model.

**Table 1 plants-12-00584-t001:** List of the 14 cultivars evaluated, along with their main fruit characteristics, ripening period, and breeding program.

	Fruit Type	Flesh Colour	Flesh Texture	Fruit Taste	Ripening Period	Breeding Programme
Carla^cov^	Peach	Yellow	Melting	Sweet	Early	PSB Producción Vegetal, Spain
Catherina^®^	Peach	Yellow	Non-melting	Equilibrate	Medium	L. Hough, USA
O’Henry	Peach	Yellow	Melting	Equilibrate	Late	G. Merrill, USA
Sweet Dream^cov^	Peach	Yellow	Melting	Very sweet	Medium	Zaiger Genetics, USA
Elegant Lady	Peach	Yellow	Melting	Sour	Medium	G. Merrill, USA
Patty^®^	Peach	White	Melting	Sour	Early	Zaiger Genetics, USA
Sweetregal^cov^	Peach	White	Melting	Very sweet	Medium	A. & L. Maillard, France
Gladys^®^	Peach	White	Melting	Sour	Late	Zaiger Genetics, USA
Sweet Cap^®^	Flat Peach	White	Melting	Very sweet	Medium	A. & L. Maillard, France
Flatstar^cov^	Flat Peach	White	Melting	Sweet	Medium	ASF, France
BigTop^®^	Nectarine	Yellow	Melting	Sweet	Medium	Zaiger Genetics, USA
Venus	Nectarine	Yellow	Melting	Sour	Medium	CREA, Italy
Nectarperf^cov^	Nectarine	White	Melting	Sweet	Late	A. & L. Maillard, France
Emeraude^®^	Nectarine	White	Melting	Sweet	Medium	R. Monteux-Caillet, France

**Table 2 plants-12-00584-t002:** Geographical location, tree density and training system in 11 locations of the EUFRIN peach trials. The sites are presented from the northern to the southern latitude.

Country	Location	Latitude	Longitude	Tree Density	Training System
Romania	Bucharest	44.4	26.1	1667	Vertical axe
Italy	Cuneo	44.4	7.5	966	Bi-axis *
Italy	Forlì	44.2	12.0	741	Open vase
Italy	Tebano	44.1	12.2	635	Open vase
France	Bellegarde	43.8	4.5	556	Double Y
Italy	Rome	41.9	12.5	741	Open vase
Spain	Zaragoza	41.7	−0.9	571	Open vase
Spain	Lleida	41.6	0.6	667	Catalan vase
Greece	Naoussa	40.6	22.1	571	Open vase
Spain	Murcia−Yéchar	38.1	−1.4	800	Open vase
Spain	Murcia−Torre Pacheco	37.8	−1.0	850	Open vase

* Hedgerow training system with two primary vertical scaffolds per tree.

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
