# Peer review of "Impact of Chill and Heat Exposures under Diverse Climatic Conditions on Peach and Nectarine Flowering Phenology"

_plants, 2023, doi:10.3390/plants12030584_

Round 1

Reviewer 1 Report

The manuscript describes the response of 14 peach/nectarine cultivars to climate change in terms of chill and heat accumulation. The topic is relevant and the general approach is finely designed and involves several research units. There are some issues regarding the literature considered and cited and the statistical tests used to separate means. All these suggestions and other minor comments are marked in the attached file and must be considered before publication.

Author Response

Reply: We thank the reviewer 1 for giving us many insightful comments that were incorporated in our revised version. The only suggestions not included were:

  • In Line 52, there was a suggestion to cite the study by Marra and Motisi. We are not sure which is the study suggested: a) Imperiale, V., et al. Estimation of chilling and heat requirements of six sweet cherry (Prunus avium L.) cultivars. In: I International Symposium on Reproductive Biology of Fruit Tree Species 1342. 2021. S. 115-122. – This paper is on sweet cherries, so not fully applicable here. b) Caruso, T., et al. The use of phenoclimatic models to characterize environments for chilling and heat requirements of deciduous fruit trees: methodological and initial results. Adv. Hort Sci., 6 (1992): 65-73. – This study presents calculations of chill and heat with various models, for a number of locations, not related to phenological observations still not applicable here. Instead, here we cited the study by Fadón, E.; Herrera, S.; Guerrero, B.I.; Guerra, M.E.; Rodrigo, J. Chilling and Heat Requirements of Temperate Stone Fruit Trees (Prunus sp.). Agronomy 2020; 10; 409. The above study is a literature review on chilling requirements of various peach cultivars, thus we believe that it is mostly appropriate.
  • In Line 149 It was suggested to cite the study by Marra, F. P., et al. Thermal time requirement and harvest time forecast for peach cultivars with different fruit development periods. In: V International Peach Symposium 592. 2001. S. 523-529. – The beta model described here was proposed for quantifying heat during the growing season, not during the dormant season.
  • Including quote marks for all names of cultivars was chosen so that it would be uniform in the manuscript. Anyway, we leave this subject to the editor suggestion.

Reviewer 2 Report

In the manuscript titled " Impact of chill and heat exposures in diverse climatic conditions on peach and nectarine flowering phenology" authors studied the resilience of peach and nectarine cultivars to variable chill and exposures and to assess the effectiveness of different chilling accumulation models to predict blooming dates, when applied to different pedoclimatic conditions and years. 

The manuscript is well- written and I feel readers of the journal and scholars will find the research interesting. The results have been presented well the graphs and tables are self-explanatory.

My suggestion to the authors is to explain about the importance of their study in Introduction and in the discussion part of the manuscript with some more examples. Authors can also explain how extreme temperatures affect the peach trees reproductively in Introduction giving readers an idea about the climate change crisis. 

Author Response

Reply: The introduction and conclusion parts were rewritten to properly emphasize the significance of the study. We agree with the reviewer that extreme temperatures may have an impact on several crucial parameters of peach trees, such as fruit size and color, poor fruit set and production rates, and fruit ripening time, however these factors are outside the scope of this paper. On the other hand, the experimental tool we have established within the EUFRIN network, which includes a similar varietal pool in as many as 11 climatically very different sites, as well as the gathering of additional thermal and agronomic data, will in the future also allow us to study the responses of peach to more and more frequent thermal extremes.

Reviewer 3 Report

The authors discuss chilling and heat accumulation, dormancy release and blooming of fourteen peach and nectarine cultivars in eleven locations (five countries) in Europe for three years. Although the topic is interesting, the experimental locations are too few to provide a representative view of the peach and nectarine flowering phenology in each country and as a whole.

In addition, the different training systems among countries, even in the same country, combining with different tree densities (Table 1)  and local standards decrease the robustness of the experiment, increasing thus bias.

Finally, the results presented by the authors are doubtful resulting in a manuscript which in its current version is not suitable for the journal.

Author Response

Reply:  The network of cultivar testing trials we established under EUFRIN is indeed a unique tool in the EU to study the influence of climatic conditions on peach phenology and adaptation, since it includes 11 sites across a wide gradient of climatic conditions, and these locations span a wide range of chill and heat accumulation during the chilling and forcing periods preceding bloom. The harmonized multi-site data sets allow assessments of Genotype x Environment interactions for traits related to bloom timing, productivity and fruit quality. Results from each country may be representative only of the region of the experimental site and by no means of the whole country (which we did not claim they are). We are not aware of any evidence indicating that management practices, training systems or tree densities affect chilling and heat requirements or flowering times in a meaningful way. We are a large consortium of authors, and most of us have been working on tree phenology for many years, so we would be surprised if we missed any major results on this issue. If the reviewer has access to any such evidence, we’d therefore appreciate if this could be shared with us, so we can compose a meaningful reply.

Reviewer 4 Report

Peach is one of the most important temperate fruit species. To grow successfully, we need to know the physiological processes of peach trees, their environmental needs, and their responses to changes in environmental factors. The changed environmental conditions can result effects with great economic importance as well. The research results presented in the article provide a lot of valuable information in this research area. I support the publication of the article.

Author Response

We thank the reviewer for their comments.

Reviewer 5 Report

Drogoudi et al., in a manuscript entitled "Impact of chill and heat exposures in diverse climatic conditions on peach and nectarine flowering phenology", presented the results of many years of multi-site research. The authors mentioned that one of the research goals was to "assess the effectiveness of the 'Tabuenca test' and different chilling accumulation models to predict blooming dates when applied to different pedoclimatic conditions and years". But there is no information in the manuscript about what methods were used to achieve the goal, and what results were achieved, and there is no information about this in conclusion. Nothing is known about the Tabuenca test. However, in the Results and Discussion chapter, the authors mainly presented the results of their research with a small discussion, which is the most valuable part of new articles. 

Due to my lack of understanding of the methods used by the authors in achieving the goals and results achieved, and the insufficient discussion, I do not recommend the manuscript for publication.

Author Response

Reply: We have proceeded in a major revision on the data presentation, statistical analyses and writing so that we believe that results and discussion are easier to understand and communicate by the reader.

Round 2

Reviewer 3 Report

Reviewer comments for the manuscript titled ''Impact of chill and heat exposures in diverse climatic conditions on peach and nectarine flowering phenology'' (ID: plants-2104047 - revised version)

 The manuscript is a new revised version and discuss chilling and heat accumulation, dormancy release and blooming of fourteen peach and nectarine cultivars in eleven locations (five countries) in Europe for three years.

The authors did a good job in the revision of the manuscript and are to be congratulated for this. Taking into account both my previous review and the response of the authors to my concerns as well as the current revised version of the manuscript, the authors must clearly state in the manuscript that they 'are not aware of any evidence indicating that management practices, training systems or tree densities affect chilling and heat requirements or flowering times in a meaningful way' (from authors reply), in order to strengthen their results. In this regard, the fact that the peach and nectarine plants are deciduous species may play a role.

In addition, a table (or more tables) with the results of the analysis of the variance (at least degrees of freedom, mean squares, variance ratios and levels of significance, if applicable, for each factor, their interaction and residuals) would enhance the clarity of the manuscript.

Author Response

-We would have appreciated evidence to support the reviewer’s argument that site-specific management or cultivation system differences affect bloom dates. Still, we added a comment on the differences in the experimental protocol established in different locations. Please find below the text insertion presented below in red letters:

‘Further reasons for inaccuracies may be that cultivars may differ in their apparent base temperature and in their heat requirements for floral bud break [35, 36]. To a small extent, bloom dates obtained from different locations may also be affected by differences in tree vigor caused by variation in training systems and tree densities, or by other environmental factors, such as water conditions [37, 38].

-In response to the 2nd comment made by the reviewer we believe that no further elaboration on the clarity of the manuscript is needed in the present study. We did an ANOVA test on the dates of dormancy release and bloom dates, yet perhaps this was not even really needed, because the test of a null-hypothesis of means being equal seems like a meaningless exercise. Furthermore, the “mean” in an ANOVA implies that there’s a true value, and the replicates contain errors. Here, the truth is clearly a distribution, and the individual errors are just different draws from the distribution. The replicates also aren’t independent, since many sites may be affected by correlated climatic variation. We still use an ANOVA, because this is what has commonly been done in comparable analyses, but we aren’t entirely convinced that this is an appropriate procedure here. Adding detailed results would place more emphasis on this analysis than seems justified.

Reviewer 5 Report

In Materials and methods, the authors wrote: "To estimate the date of dormancy release we relied on cultivar-specific chilling requirements, previously determined by simultaneously conducted 'Tabuenca tests' in Lleida (Spain) and Naoussa (Greece) using the Dynamic, Utah, Positive Utah and Chilling Hours models", i.e. the results presented in Figure 2 and described in the text, were also described and used in the publication entitled "Estimation of chilling and heat requirements of peach and nectarine cultivars grown under different pedoclimatic conditions" (DOI: 10.17660/ActaHortic.2022.1352 .63). I would like the authors to clarify this problem.

The chapter Results and discussion still lacks a proper discussion of the results obtained.

Author Response

We thank the reviewer for his/her comments, yet we don’t see any discrepancy between the two studies and where a further elaboration in the results and discussion section is required.

The reviewer appears to be alluding to a potential problem of duplicate publication of the same results. Yet we only make use of chilling requirements determined earlier to now produce bloom date predictions. We see no problem in building on earlier work.

In the publication entitled "Estimation of chilling and heat requirements of peach and nectarine cultivars grown under different pedoclimatic conditions" (DOI: 10.17660/ActaHortic.2022.1352 .63) we described that the Tabuenca test was done: The same cultivars in two locations were sampled from the orchards in repeated times and placed in forced condition for 10 days, so that the day that chilling was satisfied could be determined. When estimating the day that chilling was satisfied we calculated the chilling accumulation from October 1st to the day that the sampling was made, which was the chilling requirement of the specific cultivar.

In the present manuscript we write the following:

2.4. Estimated dates of dormancy release and bloom

To estimate the date of dormancy release, we relied on cultivar-specific chilling requirements, previously determined by simultaneously conducted ‘Tabuenca tests’ in Lleida (Spain) and Naoussa (Greece) using the Dynamic, Utah, Positive Utah and Chilling Hours models [13].

“We evaluated the progress of chill accumulation and final chill across eleven European sites with importance to peach cultivation. We quantified chill accumulation between the beginning of October and the end of February using four chill models (Fig. 2). Chill accumulation clearly varied between the study locations. Significant chill usually started to accumulate around the beginning of November in Bucharest and Cuneo, 15 days later in Tebano, Forlì, Lleida, Zaragoza and Bellegarde, 30 days later in Naoussa and Rome, and more than 60 days later (in January) in Murcia-YE and Murcia-TP. …’.

By using the data obtained in our previous study and the results from Fig. 2, we then estimated the day for breaking dormancy in all locations as well the estimated bloom date using the chilling and heat requirements. We see no problem with this procedure, given that we don’t duplicate any analysis and we properly cite our earlier work.

In the Results and Discussion section, we discuss what our results might signify, we evaluate our results in light of earlier studies, and we refer to related research to discuss possible error sources. If the reviewer is missing anything else, we’d be grateful for more specific pointers.

Round 3

Reviewer 3 Report

Reviewer comments for the manuscript titled ''Impact of chill and heat exposures in diverse climatic conditions on peach and nectarine flowering phenology'' (ID: plants-2104047 - revised version)

 The manuscript is a new revised version and discuss chilling and heat accumulation, dormancy release and blooming of fourteen peach and nectarine cultivars in eleven locations (five countries) in Europe for three years.

The authors failed to address satisfactorily the second point (In addition, a table...clarity of the manuscript.) I raised in my previous review of this manuscript. In addition, from a statistical point of view, one of the analyses chosen in this manuscript could be considered false as the authors themselves stated ('We did an ANOVA test...like a meaningless exercise.' and 'The replicates also aren't independent...we aren't entirely convinced that this is an appropriate procedure here.').

Author Response

Following the suggestion made by the academic editor, we replaced ‘treatments’ to sources of variation’. ‘Results on the date of dormancy release and bloom data were subjected to a two-way analysis of variance using site and cultivar as sources of variation, while observations in different years were treated as replicates.

Reviewer 5 Report

I thank the manuscript's authors for the detailed explanation of my doubts. I recommend the article for publication in the journal Plants.

Author Response

I thank the reviewer for his/her comments.